# Effect of low-intensity focused ultrasound therapy on postpartum uterine involution in puerperal women: A randomized controlled trial

**Dongmei Wei[1,2], Zhijian Wang[3], Jun Yue[4], Yueyue Chen[1,2], Jian Meng[1,2], Xiaoyu Niu [1,2]\***

1 Department of Gynecology and Obstetrics, West China Second Hospital, Sichuan University, Chengdu, China, 2 Key Laboratory of Birth Defects and Related Diseases of Women and Children (Sichuan University), Ministry of Education, Chengdu, China, 3 Department of Gynecology and Obstetrics, Southern Hospital, Southern Medical University, Guangzhou, China, 4 Department of Gynecology and Obstetrics, Sichuan Provincial People's Hospital, Chengdu, China

\* hxfeynxy@163.com

## Abstract

### Background

Short-term poor uterine involution manifests as uterine contraction weakness. This is one of the important causes of postpartum hemorrhage, posing a serious threat to the mother's life and safety. The study aims to investigate whether low-intensity focused ultrasound (LIFUS) can effectively shorten lochia duration, alleviate postpartum complications, and accelerate uterine involution compared with the sham treatment.

### Methods

A multicenter, concealed, randomized, blinded, and sham-controlled clinical trial was conducted across three medical centers involving 176 subjects, utilizing a parallel group design. Enrollment occurred between October 2019 and September 2020, with a 42-day follow-up period. Participants meeting the inclusion and exclusion criteria based on normal prenatal examinations were randomly divided into the LIFUS group or the sham operation group via computer-generated randomization. Patients in the LIFUS group received usual care with the LIFUS protocol, wherein a LIFUS signal was transmitted to the uterine site through coupling gel, or sham treatment, where no low-intensity ultrasound signal output was emitted. The primary outcome, lochia duration, was assessed via weekly telephonic follow-ups post-discharge. The involution of the uterus, measured by uterine fundus height, served as the secondary outcome.

### Results

Among the 256 subjects screened for eligibility, 176 subjects were enrolled and randomly assigned to either the LIFUS group (n = 88) or the Sham group (n = 88). Data on the height of the uterine fundus were obtained from all the patients, with 696 out of 704 measurements

**Data Availability Statement:** Please visit the following website for the clinical data used in the research. The data has been publicly processed:

http://www.medresman.org.cn/pub/cn/proj/projectshshow.aspx?proj=3256.

**Funding:** the National Key Research and DevelopmentProgram of China (No. 2021YFC2009100),Research Projects of Sichuan Science andTechnology Department (No.2023YFQ0070), Key Research Projects of SichuanScience and Technology Department (No.2023YFG0128).

**Competing interests:** The authors have declared that no competing interests exist.

(99%) successfully recorded. Overall, a statistically significant difference was noted in time to lochia termination (hazard ratio: 2.65; 95% confidence interval [CI]: 1.82–3.85; *P* < 0.001). The decline in fundal height exhibited notable discrepancies between the two groups following the second treatment session (mean difference: −1.74; 95% CI: −1.23 to −2.25; P < 0.001) and the third treatment session (mean difference: −3.26; 95% CI: −2.74 to −3.78; *P* < 0.001) after delivery. None of the subjects had any adverse reactions, such as skin damage or allergies during the treatment.

## Conclusions

This study found that LIFUS treatment can promote uterine involution and abbreviate the duration of postpartum lochia. Ultrasound emerges as a safe and effective intervention, poised to address further clinical inquiries in the domain of postpartum rehabilitation.

## 1. Introduction

Increasing maternal age has led to a notable surge in the incidence of poor uterine involution due to higher cesarean section rates and pregnancy complications. Short-term poor uterine involution is characterized by weakened uterine contraction. This is one of the primary causes of postpartum hemorrhage, significantly endangering women's post-delivery life and safety. Poor uterine consolidation during the puerperium may lead to late postpartum hemorrhage, prolonged lochia duration, and heightened risks of pelvic and recurrent endometrial infections, follicular growth, and ovarian dysfunction. All these factors could result in low fertility rate in women [1]. Traditional clinical methods to treat uterine involution might be ineffective due to various limitations; for instance, oxytocin is commonly used as a clinical treatment agent, but its receptor sensitivity is reduced and individual differences [2]. While physical therapeutic methods like electrical stimulation and infrared light therapy [3–5], have shown promise in clinical studies, but further evidence on clinical treatment is required.

Ultrasound is a mechanical wave exceeding 20 kHz, firstly utilized in medical applications due to rectilinear propagation and energy attenuation of its acoustic beams. Initially emerging in rehabilitation medicine, its therapeutic properties have expanded beyond to encompass the treatment of musculoskeletal disorders, branching into various other medical domains. However, limitations in energy attenuation hinder its effectiveness in treating deep tissues. Thus, focused ultrasound concentrates sound waves, allowing energy to accumulate and release within deeper tissues. In China, low-intensity focused ultrasound (LIFUS) is known for its high penetration and therapeutic targeting. Recently, it has been employed to expedite uterine involution after delivery, especially in China [6, 7]. Although a few randomized trials have suggested that ultrasound therapy aids the uterine involution process, limited inferences are known due to the small sample size and potential bias. To resolve uncertainties regarding the efficacy and safety of LIFUS therapy in promoting uterine involution, a multicenter, sham-controlled randomized clinical trial was conducted. This trial involved direct measurement of the fundal height, recording lochia duration and monitoring adverse events.

## 2. Materials and methods

### 2.1. Subjects

This multicenter, randomized, blinded, sham-controlled, parallel group clinical trial was conducted across three medical centers in China from December 2018 to October 2019. The

recruitment started on April 12, 2018 and concluded on August 30, 2019. The participating centers included West China Second Hospital of Sichuan University, Sichuan provincial people's hospital, and Southern Medical University Hospital. Inclusion criteria were as follows: pregnant women with 37 weeks of gestation and above, with no serious complications during prenatal checkup, and who breastfed their infants and performed exercise as prescribed by doctors after delivery. Patients with (1) placenta previa; (2) intrahepatic cholestasis of pregnancy; (3) triplets, twins, fetal macrosomia; (4) polyhydramnios; (5) history or signs of placental abruption; (6) severe pre-eclampsia; and (7) anticoagulant treatment for conditions like thrombocytopenia, were excluded from the study. Ethics approval was granted by the Ethics Committee of West China Second Hospital of Sichuan University on May 11, 2018 (approval No. Q2018003). And this trail was registered in the Chinese Clinical Trial Registry (ChiCTR2100049586). Delayed registration in 2018 was due to inadvertent oversight, and all ongoing and relevant trials for this intervention have been duly registered. All participating hospitals were duly informed and agreed to partake in the study. Additionally, written informed consent was obtained from all subjects before enrollment.

## 2.2. Study design

A random allocation (1:1) of LIFUS treatment and a deactivated device (identical in appearance and function) were provided to the intervention and control groups, respectively. An independent data manager divided the enrolled subjects into different groups stratified by the medical center according to a random sequence generated by a computer. Each random number is sealed in an envelope to conceal the sequence. The major investigators were aware of the treatment assigned to subjects, whereas other clinicians, data collectors, investigators, data analysts, and industry sponsors remained blinded to the allocation. Physicians recruited subjects, and using a random number method, one doctor divided them into two groups. The subjects in the LIFUS group received a Model-TY-200A ultrasound therapeutic device for involution of the uterus (Sichuan Taiyou Technology Co., Ltd., China). The device parameters were set as follows: operating frequency of 0.7 to 0.9 MHz, focus area of 0.07 to 0.085 cm$^2$, spatial peak time mean sound intensity less than 3 W/cm$^2$, and pulse duty cycle of 25% to 100%. The subjects were instructed to empty their bladders and lie supine on the treatment bed before each treatment session. The therapist located and marked the fundal position by palpating the abdomen. During the treatment, the ultrasound head was aligned to the fundus position, lightly pressed, and moved at a constant speed from side to side. Throughout the treatment, the subjects were asked about their feelings, specifically whether they felt the treatment was appropriate and whether they experienced no sensation or a slight warmth. If the subjects had obvious pain or a warm sensation, increasing the coupling agent dosage, accelerating the movement, and reducing the therapeutic energy were necessary. The control group received treatment identical to the LIFUS group, but without ultrasound energy output. The first treatment was occurred 6 hours after vaginal delivery and 24 hours after a cesarean section, given once daily for 30 minutes over 3 consecutive days. In addition, every subject received standardized postpartum care based on the "Guidelines of Normal Birth" [8]. All subjects received 20 U of intravenous oxytocin on the day of delivery.

At the end of the treatment, the fundal height of each subject was measured by the same operator when the subjects lied flat in a supine position with an empty bladder. The abdomen of the subjects was touched and pressed to locate the fundus and the uterine height was measured using a soft ruler (the distance from the midpoint of the superior border of the pubis symphysis to the fundus of the uterus). On 21 ± 2, 28 ± 2, 35 ± 2, and 42 ± 2 days after delivery, a telephonic follow-up was initiated to inquire about and record the duration of lochia for all

subjects. Should any discomfort arise, such as abdominal pain, bloody lochia more than menstrual flow, or odorous discharge, immediate treatment was provided. On the 42nd day post-delivery, subjects were instructed to visit the hospital to assess uterine involution. All subjects who complied with the study design criteria were assessed.

## 2.3. Evaluation criteria for efficacy and safety

The primary outcome measured was lochia termination. A significant difference in the reduction of total lochia duration between the LIFUS and control groups served as an indicator of treatment efficacy. The secondary outcome was the height of the uterine fundus after treatment. The fundal heights of the subjects in both groups were measured before the first treatment and after each treatment. The notable discrepancy in the decrease of the fundal height observed during the last treatment between the subjects of the LIFUS and control groups indicated treatment efficacy [9].

Safety evaluation encompassed an evaluation of adverse events observed during the clinical trials, including but not limited to skin damage, allergies, and abdominal wall, or abdominal pain.

## 2.4. Statistical analyses

Primary analysis was conducted on subjects as per the intention-to-treat (ITT) principle, involving all randomized patients within the ITT population included. The modified full analysis set included all randomized patients who received at least one intervention.

The R 3.6.1 statistical software was used for all analyses. The baseline statistics analyses in two groups were calculated for demographic variables, such as age, weight, and time of pregnancy. The independent sample t-test was applied if the statistical test was normally distributed; otherwise, the Mann-Whitney test was used. The frequency of rank variables was statistically described, and the chi-square test was employed to analyse the difference in the frequency of lochia termination subjects between the two groups. The Kaplan—Meier time-to-event curves was used for lochia duration in puerpera. The Cox proportional hazards model assessed the time discrepancy in lochia termination between the LIFUS and Sham groups. Additionally, the generalized estimating equations were applied to analyse the treatment effect on the fundus drop height effectively.

The missing data of the second outcome were imputed utilizing the following guideline: immediately after delivery, the fundus was normally firm, located midway between the symphysis pubis and umbilicus; within the subsequent 12 hours, it ascended to a level just above or below the umbilicus before gradually receding by approximately 1 cm per day, ultimately returning to its midpoint between the symphysis pubis [10].

## 2.5. Sample size

The PASS.15 software was used to calculate the sample size, which was informed by a prior study where the duration of lochia was 30.90±9.15 for patients treated with ultrasound therapy devices and 37.00±11.21 for patients receiving deactivated devices [11]. With a two-sided alpha level of 0.05, the statistical power was set to 90%. The number of cases in each group was calculated to be 64. Considering a statistically acceptable 20% drop-out (lost follow-up) rate and an additional 8% probability of other terminations, a total sample size of 176 patients was determined for this study.

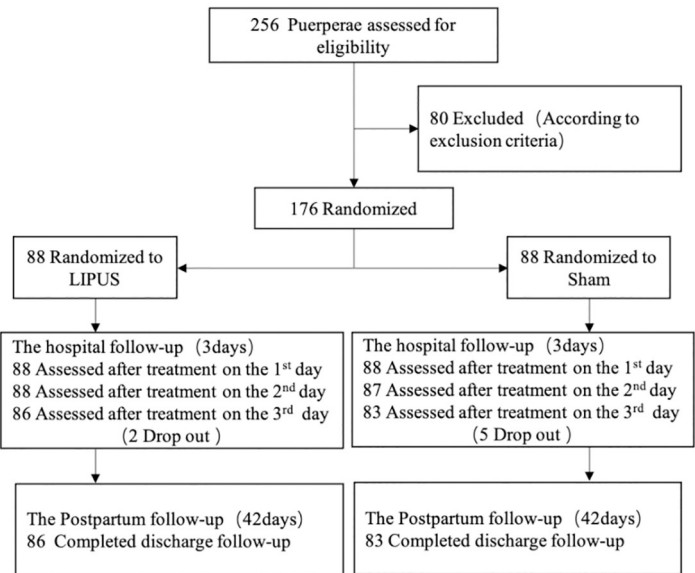

**Fig 1. Flow of the puerpera through the study of LIFUS.**

## 3. Results

### 3.1. Baseline characteristics

A total of 256 subjects were eligible for inclusion in the study. Eventually, 176 subjects were enrolled between October 2019 and March 2020 across the three centers. Seven subjects (two in LIPUS group; five in Sham group) did not adhere to the study protocol (Fig 1). Baseline data included age, number of pregnancies, number of births given, and routine blood investigations, such as HGB, HCT, RBC, WBC, PLT, and APTT. The baseline characteristics of the patients were well balanced across the two groups (Table 1).

### 3.2. Effectiveness of assessment of lochia duration

Overall, the median and quartile follow-up period was 38 (32, 41) days, with 34 (29,39) days in the LIFUS group and 40 (37,42) days in the Sham group. During the last follow-up, 70% of the subjects (119/169) experienced lochia termination of which 85% (73/86) were from the LIFUS group and 55% (46/83) from the control group. Additionally, 30% (50/169) of the subjects had no lochia termination, of which 15% (13/86) were from the LIFUS group and 45% (37/83) from the control group. A statistically significant difference in the number of subjects with lochia termination existed between the two groups ($P < 0.001$), as determined by the chi-squared test. The differences between the two groups in lochia termination were also analyzed using the Cox proportional hazards model, wherein unadjusted statistics were reported. Significant disparities were noted between the LIFUS and Sham groups (hazard ratio [HR]: 2.65; 95% confidence interval [CI]: 1.82–3.85; Table 2). Employing the log-rank test, a comparative analysis was undertaken on the Kaplan—Meier time-to-event curves representing lochia duration in the two groups. The results disclose a substantial disparity in the termination period of lochia between these groups, as depicted in Fig 2. Among the 119 subjects with lochia termination, lochia duration for vaginal delivery and cesarean section was 30.73 ± 5.59 days (n = 22) and 33.08 ± 6.04 days (n = 51) in the LIFUS group and 37 ± 4.8 days (n = 13) and 34.91 ± 5.76

**Table 1. Baseline puerperal characteristics among patients receiving LIFUS and Sham groups.**

| Variable | Total Sample (n = 176) | Sham (n = 88) | LIFUS (n = 88) | P |
|---|---|---|---|---|
| Age(years) | 31.28±4.47 | 31.09±4.51 | 31.47±4.44 | 0.58 |
| HGB (g/L) | 123.19±12.54 | 123.4±12.5 | 122.97±12.65 | 0.82 |
| HCT (%) | 28.07±16.3 | 28.13±16.39 | 28.02±16.29 | 0.97 |
| RBC ($10^{12}$/L) | 4.06±0.39 | 4.09±0.41 | 4.04±0.38 | 0.41 |
| WBC ($10^9$/L) | 9.86±6.65 | 10.26±8.92 | 9.46±2.99 | 0.42 |
| PLT ($10^9$/L) | 184.44±59.98 | 183.44±60.16 | 185.44±60.12 | 0.83 |
| APTT (sec) | 26.74±2.46 | 26.78±2.23 | 26.71±2.69 | 0.85 |
| Height of fundus before treatment (cm) | 19.41±2.64 | 19.49±2.39 | 19.34±2.88 | 0.71 |
| Number of pregnancies | | | | 0.34 |
| 1, no. (%) | 59 (33.52) | 26 (29.54) | 33 (37.5) | |
| ≥2, no. (%) | 117(66.48) | 62(70.46) | 55(62.5) | |
| Number of births | | | | 0.88 |
| 1, no. (%) | 98 (55.68) | 48 (54.54) | 50 (57) | |
| ≥2, no. (%) | 78(44.32) | 40(45.46) | 38 (43) | |

HGB: hemoglobin; HCT: hematocrit; RBC: red blood cell; WBC: white blood cell; PLT: Platelets; APTT: activated partial thromboplastin time. The independent sample t-test was applied to analyse continuous variables, and the data were presented as the mean ± standard deviation. The chi-square test was employed to analyse the frequency of rank variables, and the data were presented as n(%).

days (n = 33) in the Sham group, respectively. Comparison between cesarean section and vaginal delivery exhibited no statistical distinction in lochia duration between the two groups (Table 3).

### 3.3. Effectiveness of assessment of uterine fundus

Compared to baseline, the mean height of the uterine fundus in the LIFUS group descended to 19 (17, 20.5) cm, 17 (15, 18.5) cm, and 15 (13, 16.5) cm after the first, second, and third treatments, respectively. Whereas the Sham group exhibited a reduction to 20 (18, 21.75) cm, 18 (17, 19) cm, and 16.5 (15, 18) cm, respectively (Table 4, Fig 3).

Tables 5 and 6 summarizes the results of the generalized estimating equation for each follow-up measurement, illustrating the treatment effect on the fundus drop height effectively. According to the results of interaction effect test (Table 5), the "group * time of treatments" interaction is statistically significant (P<0.0001). The mean difference between the two groups at baseline was −0.15 cm (95% CI: −0.92–0.63). After the first treatment, the mean difference between the two groups was −0.41 cm (95% CI: −1.02–0.20). However, subsequent to the second and third treatment sessions, the mean differences were −0.85 cm (95% CI: −1.50–0.20) and −1.43 cm (95% CI: −2.14–0.73), correspondingly. These findings also revealed that the

**Table 2. Unadjusted association between LIFUS vs Sham in the overall patients.**

| | Overall (N = 169), unadjusted | | |
|---|---|---|---|
| | No. (%) | | |
| Outcome | LIFUS (n = 86) | Sham (n = 83) | HR (95% CI) |
| Lochia termination | 73 (85) | 46 (55) | 2.65 (1.82–3.85) |

Data were presented as the deviation in quantity (percentage). HR obtained from the Cox proportional hazards model.

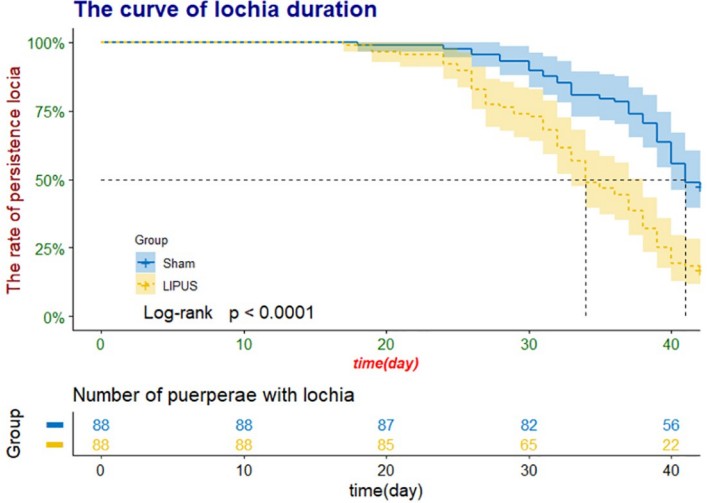

**Fig 2. Kaplan—Meier time-to-event curves for lochia duration in puerpera according to treatment with low-intensity focused ultrasound or sham device.**

**Table 3. Lochia duration of vaginal delivery vs. cesarean section.**

| lochia duration (days) | ALL | Vaginal delivery | Cesarean section | P |
|---|---|---|---|---|
| Sham | 35.5±5.54 (n = 46) | 37.0±4.8 (n = 13) | 34.91±5.76 (n = 33) | 0.17 |
| LIFUS | 32.37±5.97 (n = 73) | 30.73±5.59 (n = 22) | 33.08±6.04 (n = 51) | 0.11 |

Data were presented as the mean ± standard deviation.

P-values obtained from the independent-samples t-test.

**Table 4. Height of the fundus for each measurement.**

| Height | LIFUS | Sham | P |
|---|---|---|---|
| Baseline | 19(18, 22) | 20(18, 21) | 0.58 |
| First treatment | 19(17, 20.5) | 20(18, 21.75) | 0.07 |
| Second treatment | 17(15, 18.5) | 18(17, 19) | 0.005 |
| Third treatment | 15(13, 16.5) | 16.5(15, 18) | <0.001 |

Data were presented as the M($P_{25}$, $P_{75}$). P-values obtained from the Mann-Whitney test.

difference in height of uterine fundus descent between the LIFU group and the Sham group before and after treatment increased with the number of treatments. Furthermore, the effect of LIFUS in treating uterine fundus descent was greater than that of the Sham group. After the third treatment, the fundal height decreased to 5.04 ± 1.33 (n = 23) and 5.14 ± 1.54 (n = 63) in the LIFUS group and to 2.75 ± 1.54 (n = 24) and 2.93 ± 1.08 (n = 59) in the Sham group compared with baseline in vaginal delivery and cesarean section, respectively. There was no statistically significant difference was noted in the fundal descent height between vaginal delivery and cesarean section (Table 7).

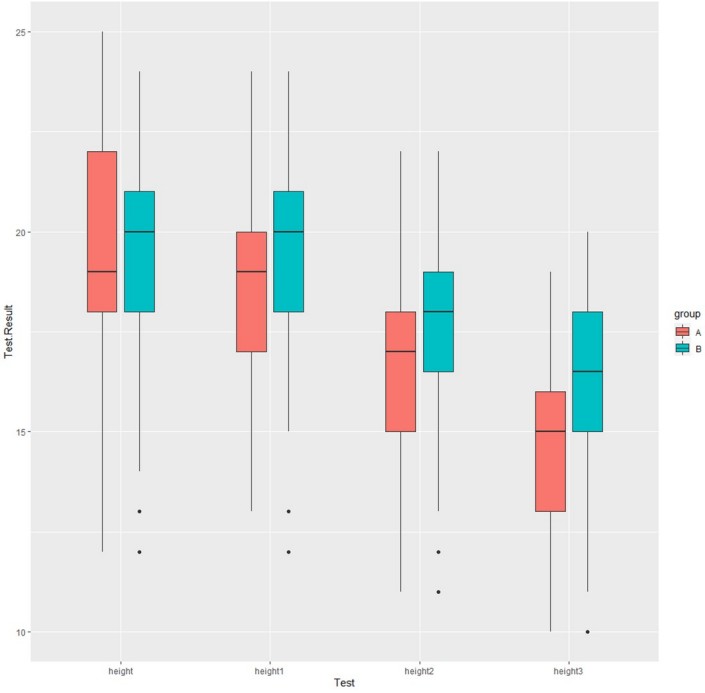

**Fig 3. Height of the fundus for each measurement.** A. LIFUS; B. Sham.

## 3.4. Safety evaluation

None of the subjects had any adverse reactions, such as skin damage or allergies during treatment. Even at the end of the study, treatment-associated adverse events were not reported in any subjects.

## 4. Discussion

The outcomes of this multicenter, randomized controlled trial revealed that the LIFUS group showed a notably greater reduction in the fundal height and exhibited a shorter duration of lochia compared to the Sham group. There was no significant disparity between cesarean section and vaginal delivery in terms of uterine descent height and lochia termination time. In this study, cesarean section and vaginal delivery did not have a significant effect on the outcomes of uterine involution in either the LIFUS or the Sham groups, possibly requiring larger sample sizes for verification. Therefore, prospective controlled studies with larger sample sizes are essential to further validate the findings of this study. The results of this study suggested that LIFUS treatment not only enhances uterine contraction but also expedites postpartum uterine involution and shortens postpartum lochia duration.

**Table 5. Results of the generalized estimating equation for interaction effect test.**

| variable | Wald chi-square | df | Sig. |
|---|---|---|---|
| Intercept | 194.109 | 1 | <0.001 |
| group | 11.372 | 1 | 0.001 |
| time of treatments | 665.481 | 3 | <0.001 |
| group * time of treatments | 24.622 | 3 | <0.001 |

**Table 6. Results of the generalized estimating equation for each measurement at follow-up.**

| Variable | Estimate | SE | Pr (>|W|) |
|---|---|---|---|
| Intercept | 19.49 | 0.25 | <0.001 |
| Group Sham | 0 | | |
| Group LIFUS | −0.15 | 0.40 | 0.714 |
| Baseline | 0 | | |
| First treatment | −0.03 | 0.26 | 0.895 |
| Second treatment | −1.74 | 0.26 | <0.001 |
| Third treatment | −3.26 | 0.27 | <0.001 |
| Group Sham * Baseline | 0 | | |
| Group Sham * First treatment | 0 | | |
| Group Sham *Second treatment | 0 | | |
| Group Sham * Third treatment | 0 | | |
| Group LIFUS * Baseline | 0 | | |
| Group LIFUS * First treatment | −0.41 | 0.31 | 0.19 |
| Group LIFUS * Second treatment | −0.85 | 0.33 | 0.009 |
| Group LIFUS * Third treatment | −1.43 | 0.36 | <0.001 |

SE: Standard error.

**Table 7. Fundus descent height of vaginal delivery vs cesarean section.**

| Fundus descent height (cm) | ALL | Vaginal delivery | Cesarean section | P |
|---|---|---|---|---|
| Sham | 2.88±1.22 (n = 83) | 2.75±1.54 (n = 24) | 2.93±1.08 (n = 59) | 0.21 |
| LIFUS | 5.12±1.48 (n = 86) | 5.04±1.33 (n = 23) | 5.14±1.54 (n = 63) | 0.86 |

Fundus descent height: the variance between the fundus height of baseline and the fundal height after the third treatment.

Data were presented as the mean ± standard deviation. P-values obtained from the independent-samples t-test.

Conventional treatments, such as postpartum oxytocin injection or uterus massaging to promote uterine contraction, were efficacious. Nevertheless, oxytocin receptor saturation was observed, wherein an increase in the dose of oxytocin failed to sustain a continuous augment in contraction effect but rather elevated the risk of toxic side effects [2]. For the case of uterus massaging, inconsistencies and irregularities in the time, speed, and strength were observed. To target this clinical dilemma, new techniques or methods are needed to attain better intervention effects.

LIFUS is a mechanical wave, which has good tissue penetration, localization, and energy deposition. This technology allows energy to transmit through surface tissue and focus on target tissue at a specific depth, triggering biological effects and inducing tissue cell changes [12]. Low-frequency and low-dose ultrasound, devoid of destructive thermogenic reactions, finds widespread application in repairing muscle injury, promoting fracture healing, and various physiotherapy fields [13, 14]. Notably, LIFUS can promote uterine contraction and reduce the duration of postpartum lochia. And its action might be attributed to the following reason: 1. The mechanical effect of ultrasound generates mechanical stimulation, particularly sensitive to uterine smooth muscle tissue at a certain frequency, eliciting contractions [15]. The mechanism by which low-intensity ultrasound promotes uterine smooth muscle contraction might be associated with changes in cell-membrane permeability to $Ca^{2+}$, thereby facilitating extracellular

fluid $Ca^{2+}$ influx and/or releasing intracellularly stored $Ca^{2+}$ [16, 17]. 2. Ultrasound may upregulate the expression of uterine endometrium and myometrium oxytocin receptors and enhance the uterine contraction response at a molecular level [18, 19]. 3. Ultrasound facilitates the repair and regeneration of the endometrium and shorten the duration of postpartum lochia by bolstering uterine contractility. Asuka Yoshii et al. [20] studied the impact of postpartum uterine contractions on mouse endometrium, demonstrating the occurrence of transient hypoxia in postpartum uteri. Concurrently, a substantial increase in the expression of vascular endothelial growth factor and transforming growth factor beta (TGF-β) was noted. In particularly, the antifibrotic factor TGF-β3 was released during the endometrial healing process. These alterations were significantly suppressed when uterine contractions were inhibited. TGF-β3 is considered as an antifibrotic molecule, and its level is the key modulator in the scarless healing of the endometrium. Uterine contractions play a crucial role in hemostasis and endometrial regeneration, leading to a process that involves the activation of macrophages, increased proliferation of endometrial cells, and upregulation of nonfibrotic growth factors.

According to this study, LIFUS is a safe, painless, and effective physical therapeutic method. Given its ability to achieve high penetration and focus, it could effectively reach the uterine and trigger uterine contractions. Moreover, the treatment process is comfortable and painless. However, the use of LIFUS necessitates mobile manipulation and accurate localization of the uterus, imposing specific physical and technical requirements on the therapists. Ultrasound is a mechanical wave within the realm of physiotherapy, serving as a safe and side-effect-free intervention for postpartum women. This study further clinically substantiated ultrasound's role in promoting contractions in the uterine smooth muscles, thereby expediting uterine involution. For postpartum women, complications such as urinary retention and gastrointestinal flatus may emerge, potentially related to the functionality of smooth muscle. Simultaneously, our research endeavors focus on exploring the prospective clinical application of LIFUS in the postpartum period, aiming to address the needs of a wider cohort of women grappling with these postpartum challenges.

The main limitation of the study was its inability to obtain the primary outcome indicators for each patient because of insufficient follow-up time. At the final follow-up, 30% of the cases still exhibited ongoing lochia. To supplement and authenticate the findings in this study, conducting a multicenter clinical trial with a large sample size is imperative. In addition, although the fundal height was measured by the same operator at each center, potential variations might have influenced outcome evaluations. Subsequent studies should prioritize standardized procedures and operations, possibly integrating ultrasonography for more accurate observations and measurements.

## 5. Conclusions

In conclusion, this study demonstrated that LIFUS was a safe and effective method for the treatment of uterine involution. By stimulating postpartum uterine contractions and curtailing lochia duration, LIFUS expedited the process of uterine involution, thereby diminishing the incidence of poor uterine involution. These compelling findings advocate for the widespread clinical adoption of LIFUS as a treatment for postpartum uterine involution.

## Supporting information

**S1 Protocol. Clinical trial protocol of ultrasound postpartum rehabilitation therapy instrument.**
(DOCX)

**S1 Checklist. CONSORT 2010 checklist of information to include when reporting a randomised trail\*.**
(PDF)

## Acknowledgments

Thanks to Sichuan Taiyou Technology Co., Ltd., China for the technical support of low intensity focused ultrasound and all the participants who contributed to this study.

## Author Contributions

**Conceptualization:** Dongmei Wei, Xiaoyu Niu.

**Data curation:** Zhijian Wang, Jun Yue, Jian Meng.

**Formal analysis:** Dongmei Wei, Jun Yue, Jian Meng.

**Funding acquisition:** Xiaoyu Niu.

**Investigation:** Zhijian Wang, Jun Yue, Yueyue Chen, Jian Meng.

**Methodology:** Dongmei Wei, Jun Yue, Yueyue Chen.

**Project administration:** Dongmei Wei, Yueyue Chen, Xiaoyu Niu.

**Resources:** Jian Meng.

**Software:** Dongmei Wei.

**Supervision:** Zhijian Wang, Yueyue Chen, Jian Meng, Xiaoyu Niu.

**Validation:** Jian Meng, Xiaoyu Niu.

**Visualization:** Zhijian Wang, Xiaoyu Niu.

**Writing – original draft:** Dongmei Wei, Zhijian Wang, Jun Yue, Yueyue Chen, Jian Meng.

**Writing – review & editing:** Dongmei Wei, Zhijian Wang, Jun Yue, Jian Meng, Xiaoyu Niu.

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
