## [Decision Letter · Decision Letter 0]

30 Aug 2023

PONE-D-23-21555

Effect of Ultrasound Therapy on Postpartum Uterine Involution: A Randomized Controlled Trial

PLOS ONE

Dear Dr. Niu,

Thank you for submitting your manuscript to PLOS ONE. After careful consideration, we feel that it has merit but does not fully meet PLOS ONE’s publication criteria as it currently stands. Therefore, we invite you to submit a revised version of the manuscript that addresses the points raised during the review process.

We look forward to receiving your revised manuscript.

Kind regards,

Ahmed Mohamed Maged, MD

Academic Editor

PLOS ONE

Journal Requirements:

2. Thank you for submitting your clinical trial to PLOS ONE and for providing the name of the registry and the registration number. The information in the registry entry suggests that your trial was registered after patient recruitment began. PLOS ONE strongly encourages authors to register all trials before recruiting the first participant in a study.

1) your reasons for your delay in registering this study (after enrolment of participants started);

2) confirmation that all related trials are registered by stating: “The authors confirm that all ongoing and related trials for this drug/intervention are registered”.

6. Please note that funding information should not appear in any section or other areas of your manuscript. We will only publish funding information present in the Funding Statement section of the online submission form. Please remove any funding-related text from the manuscript.

8. We note that the original protocol file you uploaded contains a confidentiality notice indicating that the protocol may not be shared publicly or be published. Please note, however, that the PLOS Editorial Policy requires that the original protocol be published alongside your manuscript in the event of acceptance. Please note that should your paper be accepted, all content including the protocol will be published under the Creative Commons Attribution (CC BY) 4.0 license, which means that it will be freely available online, and any third party is permitted to access, download, copy, distribute, and use these materials in any way, even commercially, with proper attribution.

Therefore, we ask that you please seek permission from the study sponsor or body imposing the restriction on sharing this document to publish this protocol under CC BY 4.0 if your work is accepted. We kindly ask that you upload a formal statement signed by an institutional representative clarifying whether you will be able to comply with this policy. Additionally, please upload a clean copy of the protocol with the confidentiality notice (and any copyrighted institutional logos or signatures) removed.

Additional Editor Comments:

Please respond to all reviewers comments one by one clearly

Reviewers' comments:

Reviewer's Responses to Questions

**Comments to the Author**

1. Is the manuscript technically sound, and do the data support the conclusions?

Reviewer #1: Partly

Reviewer #2: Partly

Reviewer #3: No

2. Has the statistical analysis been performed appropriately and rigorously? 

Reviewer #1: Yes

Reviewer #2: Yes

Reviewer #3: No

3. Have the authors made all data underlying the findings in their manuscript fully available?

Reviewer #1: Yes

Reviewer #2: Yes

Reviewer #3: Yes

4. Is the manuscript presented in an intelligible fashion and written in standard English?

Reviewer #1: Yes

Reviewer #2: No

Reviewer #3: Yes

5. Review Comments to the Author

Reviewer #1: The Ms completed by Niu Xiaoyu and co-authors mainly focused on studying the effect of Ultrasound Therapy on Postpartum Uterine Involution and the effects were principal evaluated via the duration of postpartum lochia. The data in Ms could partially support the conclusion that Ultrasound Therapy contributes to uterine involution and reduces the duration of postpartum lochia. Here I have some major concerns: 1, the functions of Ultrasound Therapy on Postpartum Uterine Involution have been reported several years ago by other groups, so pls showing your contributions detailly. 2, for the assay, you just chose one set-up parameters of ultrasound, can you tell us the underlying reasons. 3, the resolution ratio of the figures in Ms somehow is too low.

Reviewer #2: 1) Subjects

It is unclear what were the exclusion criteria. Were the following patients included or excluded?: "Patients with (1) placenta previa; (2) intrahepatic cholestasis of pregnancy; (3) triplets, twins, fetal macrosomia; (4) polyhydramnios; (5) history or signs of placental abruption; (6) severe pre-eclampsia; and (7) anticoagulant treatment, such as that for thrombocytopenia, for being included in the study."

2) Study design

The investigators blinded the treating physicians but not the participants. If all participants received standard of care treatment, what exactly was the rational for not blinding the participants to the allocated treatment (LIFUS or sham)? This should be stated and explained as this would be considered a significant study limitation.

3) Sample size calculation

Child birth and post partum haemorrhage are not uncommon phenomena and I would guess that the sample size of 176 is too small and the study is likely underpowered, even without doing a sample size calculation. It is unclear how sample size calculation was conducted and based on exactly what evidence. If it was based on a prior study, then please provide references and explanations. If there are no prior studies, then power calculation is irrelevant and the findings would be hypothesis generating.

4) The manuscript has multiple grammatical errors, typos, and inconsistent formatting. For example, variable font sizes within the text, inconsistent significant figure reporting, and grammatical errors and typos that can change or lead to misinterpretation of the text. Please follow the author guidelines for formatting the texts, figures, tables, legends, etc, and carefully proofread your manuscript in your revision.

Reviewer #3: A randomized sham-controlled clinical trial was conducted which aimed to describe lochia duration, postpartum symptoms, and uterine involution with the use of low-intensity focused ultrasound (LIFUS) compared to sham treatment. A statistically significant shorter lochia duration was observed in the intervention group compared to controls. Additionally, the height of the fundus differed significantly following the second and third treatment sessions.

Major revision:

Provide a comprehensive statistical methods section which includes all the statistical testing methods used for the analysis. Be specific.

Detailed revisions:

1- Express p-values more precisely, rather than p<0.05.

2- Abstract: For significant conclusions, state the direction of the effects.

3- Line 13 is unclear. “The doctors enroll subjects, one doctor assigns participants to subjects.”

4- Page 11, Line 5: Clarify the type of summary statistics that have been provided.

5- Page 11, Line 24: State the type of descriptive statistics that have been provided.

6- Page 12, Line 6: Clarify the type of summary statistics that have been provided.

7- Page 12: Sample size justification: The power calculation should include: (1) the estimated outcomes in each group; (2) the α (type I) error level; (3) the statistical power (or the β (type II) error level); (4) the target sample size, (5) for continuous outcomes, the standard deviation of the measurements, and (6) the statistical testing method which achieves the stated power.

8- Table 1: State the type of summary statistics provided. State the statistical testing methods used to estimate the p-values.

9- Page 12, Line 18: Provide a measure of dispersion for the median follow-up time. First and third quartiles or ranges would be appropriate.

10- Table 3: State the type of summary statistics provided.

11- Page 13, Line 2: The statistical term for average is mean.

12- Page 13, Line 12: A formal test of the interaction effect should be made. The interpretation of the results is confusing. Provide more clarity.

13- Table 5: Define “Group Plus.”

14- Table 6: State the type of summary statistics provided. Indicate the sample sizes: (1) overall, (2) for vaginal delivery, and (3) for cesarean section. State the statistical testing method(s) used to estimate the p-values.

6. PLOS authors have the option to publish the peer review history of their article (what does this mean?). If published, this will include your full peer review and any attached files.

Reviewer #1: No

Reviewer #2: No

Reviewer #3: No

---

## [Author Response · Author response to Decision Letter 0]

23 Oct 2023

Response to editor comments

Reply: We have modifid as the tamplates.

2. Thank you for submitting your clinical trial to PLOS ONE and for providing the name of the registry and the registration number. The information in the registry entry suggests that your trial was registered after patient recruitment began. PLOS ONE strongly encourages authors to register all trials before recruiting the first participant in a study.

1) your reasons for your delay in registering this study (after enrolment of participants started);

2) confirmation that all related trials are registered by stating: “The authors confirm that all ongoing and related trials for this drug/intervention are registered”.

Reply: We have added it in the Methods section in page 4 line 7-10.

Reply: We will check the Funding Information.

Reply: We have provided the URL in the data sharing.

Reply: We will linking an ORCID iD when submitting a review.

6. Please note that funding information should not appear in any section or other areas of your manuscript. We will only publish funding information present in the Funding Statement section of the online submission form. Please remove any funding-related text from the manuscript.

Reply: We have deleted the funding information.

Reply: We have deleted the Supporting Information files at the end of my manuscript.

8. We note that the original protocol file you uploaded contains a confidentiality notice indicating that the protocol may not be shared publicly or be published. Please note, however, that the PLOS Editorial Policy requires that the original protocol be published alongside your manuscript in the event of acceptance. Please note that should your paper be accepted, all content including the protocol will be published under the Creative Commons Attribution (CC BY) 4.0 license, which means that it will be freely available online, and any third party is permitted to access, download, copy, distribute, and use these materials in any way, even commercially, with proper attribution.

Reply: We have modified the files 

Therefore, we ask that you please seek permission from the study sponsor or body imposing the restriction on sharing this document to publish this protocol under CC BY 4.0 if your work is accepted. We kindly ask that you upload a formal statement signed by an institutional representative clarifying whether you will be able to comply with this policy. Additionally, please upload a clean copy of the protocol with the confidentiality notice (and any copyrighted institutional logos or signatures) removed.

Reply: A statement signed have uploaded.

Response to Reviewers

Reviewer #1: The Ms completed by Niu Xiaoyu and co-authors mainly focused on studying the effect of Ultrasound Therapy on Postpartum Uterine Involution and the effects were principal evaluated via the duration of postpartum lochia. The data in Ms could partially support the conclusion that Ultrasound Therapy contributes to uterine involution and reduces the duration of postpartum lochia. 

Here I have some major concerns: 

1) “The functions of Ultrasound Therapy on Postpartum Uterine Involution have been reported several years ago by other groups, so pls showing your contributions detailly.”

Answer to the comment: Thanks for the comment and suggestion! Previous studies have confirmed that ultrasound has the ability to promote uterine regeneration, but they were more single-centre studies and did not analyse the mode of delivery at the same time. Whereas the present study is a multicentre prospective randomised controlled single-blind clinical study, it also analysed the efficacy and safety of ultrasound therapy on uterine restoration in caesarean section versus normal delivery.

2) “for the assay, you just chose one set-up parameters of ultrasound, can you tell us the underlying reasons.”

Answer to the comment: Thanks for the comment! The parameters of the starting ultrasound treatment in this study were chosen as an appropriate treatment parameter, which we adjusted during the treatment according to the specific feelings of the woman.Therefore, the energy of ultrasound treatment in this study was not completely fixed, and adjustments were made to take into account the actual clinical maternal situation.The related supplements and discussion have been supplemented in the page4, line 27-29; page5 line1-2 in the revised manuscript.

3) “the resolution ratio of the figures in Ms somehow is too low.”

Answer to the comment: Thanks for the comment and suggestion! we have adjusted the ratio of the figures to meet the requirements.

Reviewer #2:

1) Subjects

It is unclear what were the exclusion criteria. Were the following patients included or excluded?: "Patients with (1) placenta previa; (2) intrahepatic cholestasis of pregnancy; (3) triplets, twins, fetal macrosomia; (4) polyhydramnios; (5) history or signs of placental abruption; (6) severe pre-eclampsia; and (7) anticoagulant treatment, such as that for thrombocytopenia, for being included in the study."

Answer to the comment: Thanks for the comment and suggestion! We have modified the sentence in manuscript. These patients were excluded in this study.

2) Study design

The investigators blinded the treating physicians but not the participants. If all participants received standard of care treatment, what exactly was the rational for not blinding the participants to the allocated treatment (LIFUS or sham)? This should be stated and explained as this would be considered a significant study limitation.

Answer to the comment: Thanks for the suggestion! Sorry for the lack of clarity due to writing problems, the study was blinded for subjects and other clinical workers, as well as for statistics and data collection. Have modified the sentence in manuscript at page 4 line16-18.

3) Sample size calculation

Child birth and post partum haemorrhage are not uncommon phenomena and I would guess that the sample size of 176 is too small and the study is likely underpowered, even without doing a sample size calculation. It is unclear how sample size calculation was conducted and based on exactly what evidence. If it was based on a prior study, then please provide references and explanations. If there are no prior studies, then power calculation is irrelevant and the findings would be hypothesis generating. 

Answer to the comment: Thanks for the comment and suggestion! Sample size calculation was conducted and based on a prior study, has provided in the reference. The PASS.15 software was used and the sample size was calculated based on previous studies, which was added in section 2.5. The study is a preliminary pre-study, and the sample size will be further enlarged and the follow-up will be strengthened at a later stage.

4) The manuscript has multiple grammatical errors, typos, and inconsistent formatting. For example, variable font sizes within the text, inconsistent significant figure reporting, and grammatical errors and typos that can change or lead to misinterpretation of the text. Please follow the author guidelines for formatting the texts, figures, tables, legends, etc, and carefully proofread your manuscript in your revision.

Answer to the comment: Thanks for the comment and suggestion! We will review the details of the manuscript.

Reviewer #3: A randomized sham-controlled clinical trial was conducted which aimed to describe lochia duration, postpartum symptoms, and uterine involution with the use of low-intensity focused ultrasound (LIFUS) compared to sham treatment. A statistically significant shorter lochia duration was observed in the intervention group compared to controls. Additionally, the height of the fundus differed significantly following the second and third treatment sessions.

Major revision:

Provide a comprehensive statistical methods section which includes all the statistical testing methods used for the analysis. Be specific.

Reply: We have modified it in the section 2.4.

Detailed revisions:

1- Express p-values more precisely, rather than p<0.05.

Reply: We have modified it. 

2- Abstract: For significant conclusions, state the direction of the effects.

Reply: We have modified the statement in the conclusions.

3- Line 13 is unclear. “The doctors enroll subjects, one doctor assigns participants to subjects.”

Reply: Sorry for the lack of clarity due to writing problems, We have modified the sentence in manuscript at page 4 line 18-19.

4- Page 11, Line 5: Clarify the type of summary statistics that have been provided.

Reply: We have provided the type of summary statistics in the table.

5- Page 11, Line 24: State the type of descriptive statistics that have been provided.

Reply: We have provided the type of descriptive statistics in the table.

6- Page 12, Line 6: Clarify the type of summary statistics that have been provided.

Reply: We have provided the type of descriptive statistics in the table.

7- Page 12: Sample size justification: The power calculation should include: (1) the estimated outcomes in each group; (2) the α (type I) error level; (3) the statistical power (or the β (type II) error level); (4) the target sample size, (5) for continuous outcomes, the standard deviation of the measurements, and (6) the statistical testing method which achieves the stated power.

Reply: We have tried to add relevant information as the sample size justification at section 2.5.

8- Table 1: State the type of summary statistics provided. State the statistical testing methods used to estimate the p-values.

Reply: We have added it.

9- Page 12, Line 18: Provide a measure of dispersion for the median follow-up time. First and third quartiles or ranges would be appropriate.

Reply: We have added it page8 line 8-9.

10- Table 3: State the type of summary statistics provided.

Reply: We have modified it.

11- Page 13, Line 2: The statistical term for average is mean.

Reply: We have modified it.

12-Page 13, Line 12: A formal test of the interaction effect should be made. The interpretation of the results is confusing. Provide more clarity. 

Reply: The results using gee interaction showed that the difference in height of uterine fundus descent between the LIFU group and the Sham group before and after treatment increased with the number of treatments, and that the effect of LIFU in treating uterine fundus descent was greater than that of the Sham group. Page9 line 16-19.

13- Table 5: Define “Group Plus.”

Reply: We have defined it.

13-Table 6: State the type of summary statistics provided. Indicate the sample sizes: (1) overall, (2) for vaginal delivery, and (3) for cesarean section. State the statistical testing method(s) used to estimate the p-values.

Reply: We have added it, thanks for your comments.

---

## [Decision Letter · Decision Letter 1]

29 Nov 2023

PONE-D-23-21555R1Effect of Ultrasound Therapy on Postpartum Uterine Involution: A Randomized Controlled TrialPLOS ONE

Dear Dr. Niu,

Thank you for submitting your manuscript to PLOS ONE. After careful consideration, we feel that it has merit but does not fully meet PLOS ONE’s publication criteria as it currently stands. Therefore, we invite you to submit a revised version of the manuscript that addresses the points raised during the review process.

We look forward to receiving your revised manuscript.

Kind regards,

Ahmed Mohamed Maged, MD

Academic Editor

PLOS ONE

**Additional Editor Comments:**

**Please respond to all reviewers comments one by one **. 

Reviewers' comments:

Reviewer's Responses to Questions

**Comments to the Author**

1. If the authors have adequately addressed your comments raised in a previous round of review and you feel that this manuscript is now acceptable for publication, you may indicate that here to bypass the “Comments to the Author” section, enter your conflict of interest statement in the “Confidential to Editor” section, and submit your "Accept" recommendation.

Reviewer #1: All comments have been addressed

Reviewer #3: (No Response)

2. Is the manuscript technically sound, and do the data support the conclusions?

Reviewer #1: Partly

Reviewer #3: Partly

3. Has the statistical analysis been performed appropriately and rigorously? 

Reviewer #1: Yes

Reviewer #3: No

4. Have the authors made all data underlying the findings in their manuscript fully available?

Reviewer #1: Yes

Reviewer #3: Yes

5. Is the manuscript presented in an intelligible fashion and written in standard English?

Reviewer #1: Yes

Reviewer #3: No

6. Review Comments to the Author

Reviewer #1: one important issue is the novelty of the study, just simple search the paper online, we can find many studies have done the similar job, for example, PMID: 35026964. and the introduction and discussion of paper is too simple.

Reviewer #3: Major revisions:

1- Section 2.3 Statistical analyses: Clarify the following statement. What does it mean to be "statistically described"? "The frequency of rank variables was statistically described, and the chi-square test was used to analyse the difference between the two groups."

2- The interaction effects in the GEE model, presented in table 5, do not appear to be correct. The interaction effect should be treatment (sham vs LIFUS) by session (time). The main effects are treatment and session. The full model would contain the following effects: Intercept, Treatment group, Session, and the interaction of Treatment group by Session. If the interaction effect is significant, provide an interpretation of the results, but do not test main effects because the tests for main effects are uninteresting in light of significant interactions. If interaction effects are non-significant, drop the interaction effects from the model and test the main effects. Determining which results to present when testing interactions is often a multi-step process.

Minor revisions:

1- The t-test begins with a lower case letter t.

2- Section 2.5 Sample Size: Indicate the statistical testing method which achieves 90% power. The power of 90% does not need to included twice.

3- Table 1: A) Rename the header "Full Sample" to "Total". B) Rename "Covariate" to "Variable".

4- Section 3.2: A) Indicate if the summary statistics are median and interquartile range.

B) Specify the differences of what measure between the two groups were analyzed using the Cox PH model... Cox models are typically used to predict time to an event.

6- The statistical analysis section fails to list and describe the use of the Kaplan-Meier method.

7- The following statement is vague. "Data were presented as the quantity (percention) deviation."

8- The values of the t, z and chi-square statistic can be dropped from the tables. The corresponding p-value is sufficient.

9- The standard statistical term for average is mean.

11- P-values never equal zero. Express small p-values as < 0.0001

12- Thoroughly proofread the manuscript. Many grammatical errors are present.

7. PLOS authors have the option to publish the peer review history of their article (what does this mean?). If published, this will include your full peer review and any attached files.

Reviewer #1: No

Reviewer #3: No

---

## [Author Response · Author response to Decision Letter 1]

20 Dec 2023

Reviewer #1: one important issue is the novelty of the study, just simple search the paper online, we can find many studies have done the similar job, for example, PMID: 35026964. and the introduction and discussion of paper is too simple.

Answer to the comment: Thanks for the comment and suggestion! Currently, there remains a scarcity of multicenter prospective randomized controlled trials regarding ultrasound therapy for uterine involution in the existing studies. In contrast to the mentioned study (PMID: 35026964), our research encompasses both cesarean and vaginal delivery cohorts. Clinically, we observe a higher proportion of vaginal deliveries than cesarean sections in our population, and early uterine contractions are more critical in the case of cesarean deliveries. Hence, our study holds applicability across a wider spectrum of the population. Thank you once again for the valuable suggestion, which will significantly contribute to furthering our research of this technique's clinical applications. Subsequently, we plan to conduct a larger-scale multicenter study to substantiate our findings and explore potential additional benefits for postpartum women, such as cesarean scar softening, reduction of stretch marks, and management of postpartum urinary retention. We have added some content in the introduction and discussion of paper with high light.

Reviewer #3: Major revisions:

1- Section 2.3 Statistical analyses: Clarify the following statement. What does it mean to be "statistically described"? "The frequency of rank variables was statistically described, and the chi-square test was used to analyse the difference between the two groups."

Answer to the comment: Thanks for the comment and suggestion! This is an expression error. It refers to the statistical analysis of baseline characteristics in two groups of parturients. The sentence has been modified and can be found highlighted in Section 2.4.

2- The interaction effects in the GEE model, presented in table 5, do not appear to be correct. The interaction effect should be treatment (sham vs LIFUS) by session (time). The main effects are treatment and session. The full model would contain the following effects: Intercept, Treatment group, Session, and the interaction of Treatment group by Session. If the interaction effect is significant, provide an interpretation of the results, but do not test main effects because the tests for main effects are uninteresting in light of significant interactions. If interaction effects are non-significant, drop the interaction effects from the model and test the main effects. Determining which results to present when testing interactions is often a multi-step process.

Answer to the comment: Thanks for the comment and suggestion! We conducted interaction effect tests before introducing the interaction terms, and the results are shown in Table 5(A). According to the results of the interaction effect tests, there is a statistically significant interaction effect between 'group*time' (P<0.0001), hence the incorporation of the interaction terms in Table 5(B). Additionally, we provided supplementary information for Table 5. The interaction results also indicate that an increase in the frequency of LIFUS treatments contributes more significantly to the descent of the uterine fundus compared to the sham treatment group.

Minor revisions:

1- The t-test begins with a lower case letter t.

Answer to the comment: We have modified it in this manuscript.

2- Section 2.5 Sample Size: Indicate the statistical testing method which achieves 90% power. The power of 90% does not need to included twice.

Answer to the comment: Thanks for the comment and suggestion! We have modified the sentence in section 2.5.

3- Table 1: A) Rename the header "Full Sample" to "Total". B) Rename "Covariate" to "Variable".

Answer to the comment: Thanks for the comment and suggestion! We have modified all in the table 1.

4- Section 3.2: A) Indicate if the summary statistics are median and interquartile range.

B) Specify the differences of what measure between the two groups were analyzed using the Cox PH model... Cox models are typically used to predict time to an event.

Answer to the comment:（A) Thanks for the feedback, we have made the necessary revisions. (B) In Section 3.2, the application of the Cox model to analyze the differences in lochia termination between the two groups is mentioned. 'Time' refers to the follow-up duration until lochia termination, "event" indicates the occurrence of lochia termination, with the independent variable being "group," enabling the construction of a Cox model that generated the results in Table 2.

6- The statistical analysis section fails to list and describe the use of the Kaplan-Meier method.

Answer to the comment: Thanks for the comment! We have added the Kaplan-Meier method in section 2.4.

7- The following statement is vague. "Data were presented as the quantity (percention) deviation."

Answer to the comment: Thanks for the comment! We have modified the sentence.

8- The values of the t, z and chi-square statistic can be dropped from the tables. The corresponding p-value is sufficient.

Answer to the comment: Thanks for the suggestion! We have deleted the statistical values in all tables.

9- The standard statistical term for average is mean.

Answer to the comment: we have changed it to mean.

11- P-values never equal zero. Express small p-values as < 0.0001

Answer to the comment: Thanks for the suggestion! We have modified the p-value.

12- Thoroughly proofread the manuscript. Many grammatical errors are present.

Answer to the comment: Thanks for the feedback! We’ve proofread the grammatical errors again, please point out if there are still any remaining mistakes.

---

## [Decision Letter · Decision Letter 2]

3 Jan 2024

PONE-D-23-21555R2Effect of Ultrasound Therapy on Postpartum Uterine Involution: A Randomized Controlled TrialPLOS ONE

Dear Dr. Niu,

Thank you for submitting your manuscript to PLOS ONE. After careful consideration, we feel that it has merit but does not fully meet PLOS ONE’s publication criteria as it currently stands. Therefore, we invite you to submit a revised version of the manuscript that addresses the points raised during the review process.

We look forward to receiving your revised manuscript.

Kind regards,

Ahmed Mohamed Maged, MD

Academic Editor

PLOS ONE

**Additional Editor Comments:**

Please respond to all reviewers comments

Reviewers' comments:

Reviewer's Responses to Questions

**Comments to the Author**

1. If the authors have adequately addressed your comments raised in a previous round of review and you feel that this manuscript is now acceptable for publication, you may indicate that here to bypass the “Comments to the Author” section, enter your conflict of interest statement in the “Confidential to Editor” section, and submit your "Accept" recommendation.

Reviewer #3: (No Response)

Reviewer #4: All comments have been addressed

2. Is the manuscript technically sound, and do the data support the conclusions?

Reviewer #3: Yes

Reviewer #4: Yes

3. Has the statistical analysis been performed appropriately and rigorously? 

Reviewer #3: No

Reviewer #4: Yes

4. Have the authors made all data underlying the findings in their manuscript fully available?

Reviewer #3: Yes

Reviewer #4: Yes

5. Is the manuscript presented in an intelligible fashion and written in standard English?

Reviewer #3: No

Reviewer #4: Yes

6. Review Comments to the Author

Reviewer #3: Minor revisions:

2.4 Statistical Analysis section:

1- Line 22: The term t-test begins with a lower case t.

2- Line 24 still contains the vague phase of "statistically described". Please list the statistical methods to provide more clarity.

3- Line 25: The chi-square test was used to analyze the difference in what between the two groups?

4- Line 26 is unclear due to the repeated word "time. "The Cox proportional hazards model was used to analyze the time difference in time to lochia termination between the LIFUS and sham groups."

2.5 Sample Size

1- Line 10: Grammatical error: Replace "calculated" with "calculate".

2- Line 11: Explain the term "end time".

3- State the statistical testing method which achieves 90% power. Remove "90% power" at either line 13 or 16.

4- The value of the t-test can be removed from table 1 and replace "full sample" with "total".

3.2 Effectiveness.... section

5- Line 13: State precisely which differences are being compared. Same comment for the sentence that follows.

6- Lines 19-20: No statistical differences in what between the two groups?

7- Table 2: Kaplan-Meier curves are not compared with t-tests. Use an appropriate test for comparing Kaplan-Meier curves.

8- The standard statistical term for average is mean.

9- P-values never equal zero, express small p-values as p< 0.001, even in summary tables.

Reviewer #4: Please reformat the title using PICO pattern ( clarify the population included and outcomes

Add reference for sample size calculation

Add BMI to table 1

page 10 line 30 replace included by excluded

7. PLOS authors have the option to publish the peer review history of their article (what does this mean?). If published, this will include your full peer review and any attached files.

Reviewer #3: No

Reviewer #4: No

---

## [Author Response · Author response to Decision Letter 2]

17 Jan 2024

Review Comments to the Author

Reviewer #3: Minor revisions:

2.4 Statistical Analysis section:

1- Line 22: The term t-test begins with a lower case t.

Answer to the comment: We have modified it in this manuscript.

2- Line 24 still contains the vague phase of "statistically described". Please list the statistical methods to provide more clarity.

Answer to the comment: Thanks for the comment and suggestion! We have modified the sentences.

3- Line 25: The chi-square test was used to analyze the difference in what between the two groups?

Answer to the comment: We have added relevant content.

4- Line 26 is unclear due to the repeated word "time. "The Cox proportional hazards model was used to analyze the time difference in time to lochia termination between the LIFUS and sham groups."

Answer to the comment: We have revised the sentence.

2.5 Sample Size

1- Line 10: Grammatical error: Replace "calculated" with "calculate".

Answer to the comment: Thanks for the comment, we have rectified the mistake. 

2- Line 11: Explain the term "end time".

Answer to the comment: Thanks for the comment, we have modified it. 

3- State the statistical testing method which achieves 90% power. Remove "90% power" at either line 13 or 16.

Answer to the comment: Thanks for the comment and suggestion! We have modified the sentence in section 2.5.

4- The value of the t-test can be removed from table 1 and replace "full sample" with "total".

Answer to the comment: Thanks for the comment and suggestion! We have revised all in the table 1.

3.2 Effectiveness.... section

5- Line 13: State precisely which differences are being compared. Same comment for the sentence that follows.

Answer to the comment: Thanks for the comment and suggestion! We have added it. 

6- Lines 19-20: No statistical differences in what between the two groups?

Answer to the comment: Thanks for the comment, we have modified the sentence.

7- Table 2: Kaplan-Meier curves are not compared with t-tests. Use an appropriate test for comparing Kaplan-Meier curves.

Answer to the comment: In Figure 2, we examine the disparity in follow-up period for lochia termination between two groups through Kaplan-Meier curves, utilizing the log-rank test for statistical assessment. Meanwhile, Table 2 investigates the variance in lochia termination between these groups using Cox regression and applies the t-test to assess the significance of regression coefficients.

8- The standard statistical term for average is mean.

Answer to the comment: Thanks for the comment, we have rectified the mistake.

9- P-values never equal zero, express small p-values as p< 0.001, even in summary tables.

Answer to the comment: Thanks for the suggestion! We have modified the p-value.

Reviewer #4: Please reformat the title using PICO pattern ( clarify the population included and outcomes

Answer to the comment: Thanks for the suggestion! We have modified the title.

Add reference for sample size calculation

Answer to the comment: Thanks for the suggestion! The sample size calculation is based on the methodology outlined in reference 11.

Add BMI to table 1

Answer to the comment: Thanks for the suggestion! We have added BMI to table 1.

Answer to the comment: Thanks for the suggestion! We sincerely regret the omission of BMI parameter collection in the case report form during our study. In future research endeavors, we pledge to diligently include this crucial data in our investigations.

page 10 line 30 replace included by excluded

Answer to the comment: We have modified it.

---

## [Decision Letter · Decision Letter 3]

25 Mar 2024

Effect of Low-intensity Focused Ultrasound Therapy on Postpartum Uterine Involution in Puerperal Women: A Randomized Controlled Trial

PONE-D-23-21555R3

Dear Dr. Niu,

We’re pleased to inform you that your manuscript has been judged scientifically suitable for publication and will be formally accepted for publication once it meets all outstanding technical requirements.

Kind regards,

Ahmed Mohamed Maged, MD

Academic Editor

PLOS ONE

Additional Editor Comments (optional):

Reviewers' comments:

Reviewer's Responses to Questions

**Comments to the Author**

1. If the authors have adequately addressed your comments raised in a previous round of review and you feel that this manuscript is now acceptable for publication, you may indicate that here to bypass the “Comments to the Author” section, enter your conflict of interest statement in the “Confidential to Editor” section, and submit your "Accept" recommendation.

Reviewer #3: (No Response)

Reviewer #5: (No Response)

2. Is the manuscript technically sound, and do the data support the conclusions?

Reviewer #3: Yes

Reviewer #5: Yes

3. Has the statistical analysis been performed appropriately and rigorously? 

Reviewer #3: Yes

Reviewer #5: Yes

4. Have the authors made all data underlying the findings in their manuscript fully available?

Reviewer #3: Yes

Reviewer #5: Yes

5. Is the manuscript presented in an intelligible fashion and written in standard English?

Reviewer #3: Yes

Reviewer #5: Yes

6. Review Comments to the Author

Reviewer #3: Minor revisions:

1- Line 6: Grammatical error: Insert "the" between in and two.

2- Line 8: Remove the phase, "was statistically described" and provide more clarity in this sentence.

3- State the statistical testing method which achieves 90% power. Perhaps it is the t-test.

Reviewer #5: Overall, the paper is coherently developed. However, there are some suggestions.

Abstract:

- Line 23 – Keywords: - Capitalize the words.

- Replace the semicolon with a comma.

Introduction:

- Well structured.

- Line 7: add a comma after the word ‘’reduced’’.

Methods:

Study design:

- Line 16: Replace the word ‘’were’’ with the word ‘’was’’.

Discussion:

- Well structured.

7. PLOS authors have the option to publish the peer review history of their article (what does this mean?). If published, this will include your full peer review and any attached files.

Reviewer #3: No

Reviewer #5: No
